# Deoxynivalenol Induces Caspase-8-Mediated Apoptosis through the Mitochondrial Pathway in Hippocampal Nerve Cells of Piglet

**DOI:** 10.3390/toxins13020073

**Published:** 2021-01-20

**Authors:** Li Cao, Yunjing Jiang, Lei Zhu, Wei Xu, Xiaoyan Chu, Yafei Zhang, Sajid Ur Rahman, Shibin Feng, Yu Li, Jinjie Wu, Xichun Wang

**Affiliations:** College of Animal Science and Technology, Anhui Agricultural University, Hefei 230036, China; caoli@ahau.edu.cn (L.C.); jiangyunjing001@126.com (Y.J.); zhuleizl@ahau.edu.cn (L.Z.); xuwei03@zhengbang.com (W.X.); chuxiaoyan2361@163.com (X.C.); zhangyafeivet@163.com (Y.Z.); dr_sajid226@yahoo.com (S.U.R.); fengshibin@ahau.edu.cn (S.F.); lydhy2014@ahau.edu.cn (Y.L.)

**Keywords:** deoxynivalenol, apoptosis, mitochondrial pathway, piglet hippocampal nerve cells

## Abstract

Deoxynivalenol (DON) is a common trichothecene mycotoxin found worldwide. DON has broad toxicity towards animals and humans. However, the mechanism of DON-induced neurotoxicity in vitro has not been fully understood. This study investigated the hypothesis that DON toxicity in neurons occurs via the mitochondrial apoptotic pathway. Using piglet hippocampal nerve cells (PHNCs), we evaluated the effects of different concentrations of DON on typical indicators of apoptosis. The obtained results demonstrated that DON treatment inhibited PHNC proliferation and led to morphological, biochemical, and transcriptional changes consistent with apoptosis, including decreased mitochondrial membrane potential, mitochondrial release of cytochrome C (CYCS) and apoptosis inducing factor (AIF), and increased abundance of active cleaved-caspase-9 and cleaved-caspase-3. Increasing concentrations of DON led to decreased B-cell lymphoma-2 (Bcl-2) expression and increased expression of BCL2-associated X (Bax) and B-cell lymphoma-2 homology 3 interacting domain death agonist (Bid), which in turn increased transcriptional activity of the transcription factors AIF and P53 (a tumor suppressor gene, promotes apoptosis). The addition of a caspase-8 inhibitor abrogated these effects. These results reveal that DON induces apoptosis in PHNCs via the mitochondrial apoptosis pathway, and caspase-8 is shown to play an important role during apoptosis regulation.

## 1. Introduction

Deoxynivalenol (DON) is a trichothecene mycotoxin mostly produced by *Fusarium graminearum* and *F. culmorum*. Trichothecene is the most common toxin found in numerous foods and agricultural products, such as corn, maize, wheat, barley, and oats [1,2]. In many countries, DON contamination in foods and animal feeds has been a ubiquitous problem that potentially causes poisoning and doing great harm to humans and animals [3,4,5]. DON toxicity has broad mechanisms in different species. Alternations in pharmacokinetic properties (i.e., absorption, distribution, metabolism, and excretion) of DON in different animal species might account for its varied sensitivities. In animal models, the most common consequences of extended nutritional exposure to DON are reduced weight gain, anorexia, and malnutrition, with different susceptibilities observed among different species [6,7]. Fusarium toxin contamination levels are the highest among cereals [8]. DON contamination is the highest in cereals [8] and often occurs with other mycotoxins [9]. Nevertheless, the toxic effects of DON on human beings have attracted more and more attention worldwide [10] and is therefore of significant importance to veterinary medicine, plant breeding, pathology, and public health.

DON toxicity is dose- and occurrence-dependent both in humans and in animals, and exerts a range of responses including neurotoxicity, cytotoxicity, immunotoxicity, carcinogenesis, and teratogenicity [11,12,13,14,15]. Two well-known neurologic consequences of DON have attracted interest in its neuro-pharmacologic properties. DON causes anorexia at low dietary concentrations, while at higher doses, induces vomiting [16]. DON impairment of weight gain is strongly associated with reduced food intake, which may occur through interference with the intestinal motility and eating desires. These effects likely derive from dysregulation of neuroendocrine signaling and growth hormone signaling within the enteric and central nervous systems [17,18]. Emesis associated with DON is thought to be related to its effect on serotonergic signaling [19].

Despite extensive studies on DON’s broad toxicity, limited information is available on its neurotoxicity. Our previous publications have suggested that DON exposures can affect the cerebral lipid peroxidation, neurotransmitters secretion, and the balance of calcium homeostasis in chicks [13]. DON acts through the calcium / calmodulin / calmodulin-dependent protein kinase II (Ca^2+^/CaM/CaMKII) signaling pathway to influence cerebral lipid peroxidation and neurotransmitter levels in piglets [20]. Currently, most research focuses on the neurotoxic effect of DON on piglets, but its mechanism involving nerve cells is unclear. Apoptosis is an active and orderly process in which the stimulating factors inside and outside the cells of the normal body are activated and regulated by strict regulatory signals [21]. Mitochondria are one of the most important cellular organelles for cellular energy production and survival, and the mitochondrial pathway is crucial in intracellular apoptosis signaling cascade. The apoptosis caused by the mitochondrial pathway is mainly caused by the changes of protein, gene, and mitochondrial membrane permeability caused by the stimulation of the apoptosis signal [22]. Pig is considered the most sensitive species to the toxicity of DON. Therefore, we used piglet hippocampal nerve cells (PHNCs) as a neuronal cell model, focusing on mechanisms of DON-induced apoptosis and mitochondrial signaling.

## 2. Results

### 2.1. DON Induces Apoptotic Nuclear Changes in PHNCs

Laser confocal microscopy was used to detect nuclear features (Figure 1). Normal nuclei appear uniform blue under laser confocal microscopy (Figure 1A), while apoptotic nuclei appear small and bright blue. When the concentration of DON is higher than 125 ng/mL, the morphology of the nucleus showed dose-dependent deformation with higher DON concentration, and the nuclei showed not uniformly stained (Figure 1B–F).

### 2.2. DON Significantly Increases Rate of Apoptosis in PHNCs

Cell apoptosis was investigated by flow cytometry after 24 h of exposure to different DON concentrations (0, 125, 250, 500, 1000, or 2000 ng/mL). The proportion of cell apoptosis is shown in Figure 2. Compared to untreated (control) PHNCs, the apoptotic rates for PHNCs in DON treatment groups increased significantly (*p* < 0.01). The apoptotic rate increased in a dose-dependent way between 125–1000 ng/mL, while the apoptotic rate of the 2000 ng/mL DON treatment group was lower than at 1000 ng/mL DON. Therefore, 1000 ng/mL of DON was used as the optimal concentration in subsequent trials for the addition of 10 μM (Figure 2D) caspase-8 inhibitor such as (Z-IETD-FMK, ZIF).

The influence of DON on feasibility of PHNCs after 24 h incubation was examined via the cell counting kit-8 (CCK-8) cell viability assay kit. The results showed a dose-dependent decrease in cell viability from 0 to 2000 ng/mL DON (Figure 2C): compared with the untreated (control) group, the viability of DON-treated cells was significantly decreased (*p* < 0.01), with minimal viability observed for cells dosed with 2000 ng/mL of DON.

### 2.3. DON Reduces Mitochondrial Membrane Potential

Mitochondrial membrane potential (MMP) was evaluated using flow cytometry after 24 h of DON exposure at different concentrations (0, 125, 250, 500, 1000, or 2000 ng/mL; ZIF; 1000 ng/mL + ZIF). As shown in Figure 3, the MMP of DON-treated PHNCs decreased significantly with increasing concentrations of DON, compared to untreated (control) cells (*p* < 0.01). The MMP was significantly increased in the 1000 ng/mL DON + ZIF treatment group compared to the 1000 ng/mL unaided DON treatment group (*p* < 0.01).

### 2.4. Influence of DON on Genes Expression Associated with Apoptosis

After PHNCs were grown with specified concentrations of DON (0, 125, 250, 500, 1000, or 2000 ng/mL), ZIF, or 1000 ng/mL DON + ZIF for 24 h, real-time PCR was used to detect mRNA expression levels of *Bcl-2*, *Bax*, and *Bid* (Figure 4). Compared with untreated (control) PHNCs, the *P53*, *Bax*, and *Bid* mRNA expression levels as well as the ratio of *Bax/Bcl-2* increased significantly (*p* < 0.01) with increasing concentrations of DON, and highest at the concentration of 1000 ng/mL. The mRNA expression levels of *Bcl-2* reduced with increasing concentrations of DON, and the effects were significant when the concentrations exceeded 500 ng/mL (*p* < 0.01). In addition, *P53*, and *Bid* mRNA expression levels as well as the *Bax/Bcl-2* proportion were significantly reduced (*p* < 0.01) in PHNCs treated with 1000 ng/mL DON + ZIF compared to treatment with 1000 ng/mL DON alone. Compared with 1000 ng/mL DON group, *Bcl-2* mRNA expression was significantly higher in 1000 ng/mL DON + ZIF group (*p* < 0.01).

### 2.5. Effects of DON on Proteins Related with Apoptosis

The relative expression of CYCS, AIF, caspase-3, caspase-9, cleaved-caspase-9, and cleaved-caspase3 (Figure 5A) in DON-treated PHNCs compared to untreated PHNCs showed that, expressions of mitochondrial AIF and CYCS (Mito AIF and Mito CYCS) were significantly decreased (*p* < 0.01) with DON concentrations between 250–1000 ng/mL, reaching the lowest at 1000 ng/mL (Figure 5B). In contrast, expression of AIF and CYCS in the cytoplasm (Cyto AIF and Cyto CYCS) was significantly increased and reached the highest level at 1000 ng/mL DON.

Cleaved-caspase-9 and cleaved-caspase-3 levels were significantly increased (*p* < 0.05) with DON concentrations between 250–1000 ng/mL, peaking at 1000 ng/mL (Figure 5B). Caspase-9 and caspase-3 expression was decreased (*p* < 0.05) with DON concentrations between 250–2000 ng/mL. Compared to PHNCs treated with 1000 ng/mL DON, Mito AIF and Mito CYCS expression in PHNCs treated with 1000 ng/mL DON + ZIF was increased, but the changes were not statistically significant. However, expression of Cyto AIF and Cyto CYCS in PHNCs treated with 1000 ng/mL DON + ZIF was significantly decreased compared to 1000 ng/mL DON group (*p* < 0.01). Expression of cleaved-caspase-9 and cleaved-caspase-3 in PHNCs treated with 1000 ng/mL DON + ZIF significantly increased (*p* < 0.01), while expression of caspase-9 and caspase-3 in PHNCs treated with 1000 ng/mL DON + ZIF showed the opposite effect.

### 2.6. Effect of DON on Caspase-3 Activity

After 24 h exposure to increasing concentrations of DON (0, 125, 250, 500, 1000, 2000 ng/mL), ZIF, or 1000 ng/mL DON + ZIF, we found that untreated (control) PHNCs showed minimal or even no expression of caspase-3, while cells treated with DON indicated increased expression of caspase-3 with higher concentrations of DON, reached the highest at 1000 ng/mL DON (Figure 6). Caspase-3 expression was significantly reduced in PHNC cells treated with 1000 ng/mL DON + ZIF, after comparison with 1000 ng/mL DON alone.

### 2.7. Effect of DON on the Transcriptional Activities of AIF and P53

We examined the relationship between mitochondrial release of the transcription factors AIF and P53 and nuclear transcription using EMSA. Transcriptional activities of AIF and P53 in PHNCs treated with DON significantly increased with higher DON concentrations (*p* < 0.01) (Figure 7). AIF and P53 transcriptional activity was significantly reduced in PHNCs treatment with 1000 ng/mL DON + ZIF, when compared to 1000 ng/mL DON alone (*p* < 0.01).

## 3. Discussion

DON-induced apoptosis has been confirmed in various cell types including gastrointestinal tract and intestinal epithelial cells. However, there is little information about the effects of DON in nerve cells, especially related to DON-induced apoptosis through the mitochondrial signaling pathway. We have shown that DON significantly inhibited piglet hippocampal nerve cells viability and promoted the release of LDH by damaging the membrane integrity of PHNCs. DON induced PHNCs apoptosis and its mechanism of action is related to the mitogen-activated protein kinase (MAPK) signal pathway [23]. However, the specific target organelle (mitochondrion) and signal transduction mechanism need to be further studied. Here, we used PHNCs to assess this mechanism (Graphical Abstract).

DON is known to be toxic to many cell types, with its significant cytotoxicity mediated primarily through induction of apoptosis [24]. The inference of apoptotic chromatin deviation with treatment of DON has been observed by fluorescence microscopy in lung fibroblasts, human proximal tubule cells, and human colon cancer cells [25,26]. Our experimental results are consistent with these observations. We demonstrated that PHNC viability decreased with increasing concentrations of DON, and that DON-treated cells showed typical ultrastructural changes consistent with apoptosis including nuclear shrinkage and dense fluorescence. DON can damage cell membranes, inhibit cell activity, and promote LDH release, which leads to apoptosis and cell death in PC12 cells [27]. A previous study demonstrated that ZEA exposure impaired pGCs growth and apoptosis via the miRNAs-mediated focal adhesion pathway [28], and Xu et al. reported that DON exposure can induce apoptosis in intestinal porcine epithelial cells (IPEC-J2) [29]. We confirmed through flow cytometry that apoptosis occurs in a DON dose-dependent fashion in PHNCs. The result is consistent with earlier observations. Overall, the above results revealed that DON could persuade apoptosis of PHNCs, suggesting that further research focusing on the mitochondrial apoptotic pathway is warranted.

In recent years, studies have shown that mitochondria are involved in almost all cell apoptosis [30]. MMP results from the uneven distribution of protons and ions across the mitochondrial membrane suggested MMP plays a significant role in the process of apoptosis, and it is thought that alterations in MMP occur in the earliest stages of apoptosis [31,32]. Once mitochondria are injured, the MMP is markedly decreased, leading to severe impairment of mitochondrial function and eventually irreversible apoptosis. Caspase-8 is known to play a crucial role in intervening Fas-persuaded apoptosis [33,34]. Once it is activated, the classic apoptotic cascade including activation of caspase-3, -6, and -7 would occur and ultimately leading to mitochondrial damages [35]. Due to its central role in apoptosis, we choose an inhibitor of caspase-8 to demonstrate the role of the mitochondrial apoptosis pathway in DON-associated toxicity. In this study, we labeled mitochondria of apoptotic cells with JC-1 and measured its fluorescence using flow cytometry to quantify changes in MMP. We found that MMP of PHNCs decreased significantly after 24 h exposure to concentrations of DON. MMP was significantly increased in PHNCs cured with 1000 ng/mL DON + ZIF compared to 1000 ng/mL alone (*p* < 0.01), suggesting that caspase-8 inhibition can prevent dissipation of the MMP. Altogether, our results confirmed that mitochondria are involved in DON-mediated cell apoptosis.

Bcl-2 protein is contained in the endoplasmic reticulum, mitochondrial membrane, and the nuclear envelope, through a region in its C-terminus [36,37,38]. Studies proposed that that Bcl-2 may act on signaling molecules and mitochondrial and nuclear pore complexes such as CYCS and apoptosis inducing factor AIF, and control cell signaling to prolong cell survival [39]. The Bcl-2/Apaf-1/caspase-9 complex is directly combined with Apaf-1 to inhibit the activation of caspase-9 by caspase-3. Bcl-2 may also regulate caspase on the mitochondrial membrane and reduce its activity, but it does not affect the activation of caspase-9 by CYCS and Apaf-1 [40]. Bid is a pro-apoptotic factor and its product tBid has the ability to induce CYCS leakage from mitochondria [41,42,43,44], without dissipation of the mitochondrial inner membrane potential. CYCS released into the cytoplasm activates downstream caspase-9, further activates the caspase-3 cascade, and eventually leads to apoptosis [45]. Bcl-2 protein can inhibit apoptosis by binding to Bid, Bim, or Bad, and Bcl-2/Bax ratio concludes whether a cell will live after receiving apoptotic signals [38]. In our study, bcl-2, CYCs, caspase 3, and caspase 9 expression declined with higher concentration of DON, with expression reaching a lowest at 1000 ng/mL. With increasing of DON concentration, bid and bax expression increased, and at 1000 ng/mL, it reached a maximum level. Variations in expression of bcl-2, CYCs, caspase 3, caspase 9, bax, and bid in PHNCs preserved with 1000 ng/mL DON + ZIF were contrary to those treated with 1000 ng/mL DON alone. Our data support the conclusion that *bcl-2*, *bax*, and *bid* act a decisive role in apoptosis in DON-treated cells. These results indicated that bcl-2, bax, bid, CYCS, AIF, caspase-9, and caspase-3 contribute to DON-triggered apoptosis in PHNCs, and CYCS is unconstrained into the cytoplasm from mitochondria when DON-triggered apoptosis occurred, and that caspase-8 inhibition via ZIF could prevent the release of CYCS from the mitochondria.

AIF is an active protein that induced apoptosis and located between the mitochondrial double membranes. AIF is free from the mitochondria into the cytoplasm after stimulation by an apoptosis signal, enters the nucleus where it facilitates DNA cleavage, which may further contribute to apoptosis [46]. In our previous study [27], we found that AIF was released into the cytoplasm from the mitochondria in cells treated with DON, and that this release was prohibited by ZIF-mediated inhibition of caspase-8. The tumor suppressor gene P53, a transcription factor central to regulation of apoptosis, possesses two functions: to repair cell damage or to induce cell apoptosis. P53 could combine with DNA and checked if DNA was damaged. After DNA damage was founded by P53, it would stimulate the expression of cyclin-dependent kinase (CDK)-interacting protein (CIP), preventing cell division and allowing for DNA repair to occur. When DNA damage was higher than P53 repair, then the expression of P53 could promote cell apoptosis [47,48]. P53 could affect cell apoptosis through inhibiting the *Bcl-2* expression level and promoting *Bax* and *Bak* expression [49]. In our study, the transcription activities of P53 were enhanced with the increasing DON concentration, which confirmed the effect of P53 in cell apoptosis.

In summary, the present study indicates that DON can induce apoptosis of PHNCs via triggering of the mitochondrial signal transduction pathway. The results in the current study are similar to previous reports that DON induced mitochondria-dependent apoptosis in porcine intestinal epithelial cells and rat anterior pituitary GH3 cells [50,51]. Our study found that DON treatment led to induction of the pro-apoptotic genes *Bax* and *Bid*, while *Bcl-2* expression was repressed. DON treatment also led to cleavage (activation) of the apoptosis-related proteins caspase-3 and caspase-9. These effects were diminished by inhibition of caspase-8. In addition, Zhu et al. also reported that zearalenone induces apoptosis and necrosis in porcine granulosa cells via a caspase-3- and caspase-9-dependent mitochondrial signaling pathway [52]. Together, these data propose that mitochondrial apoptosis might be a principal mechanism through which DON induces neurotoxicity, and provide important insights for future studies on mechanisms of DON neurotoxicity.

## 4. Materials and Methods

### 4.1. Cell Culture and Treatment

The piglet hippocampal nerve cells (PHNCs) were provided by Nanjing Keygen Biological Technology which were cultured in Dulbecco’s modified Eagle medium (DMEM) (Thermo Scientific, Grand Island, NY, USA) comprising of fetal bovine serum (FBS, 10%), (Clark Bioscience, Richmond, VA, USA), and 100 U/mL of penicillin, with 100 µg/mL streptomycin under an atmosphere of 5% CO_2_. For ultrastructural studies and apoptosis assays, PHNCs were grown in logarithmic phase, harvested, and seeded in 24-well plates (1 × 10^5^ cells/mL). The cells were incubated for 24 h and cured with ascending concentrations of DON (Sigma, St. Louis, MO) (0, 125, 250, 500, 1000, 2000 ng/mL) for 24 h, the concentrations of DON were referred in our previous research [20]. In two experimental groups, 10 µM caspase-8 inhibitor (Z-IETD-FMK, Keygen Biotech, Nanjing, China) was added (0, 1000 ng/mL DON) 30 min prior to DON treatment to assess the mitochondrial caspase-8-mediated apoptosis pathway in DON exposure response. Cells were collected for evaluation of Cytochrome C, AIF, caspases, and *bcl-2* family members.

### 4.2. Viability Assay

CCK-8 cell viability assay kit (Dojindo Molecular Technologies, Inc., Tokyo, Japan) was used to detect cell viability rendering to the protocol of the manufacturer. Briefly, 1 × 10^5^ cells/mL in a volume of 100 µL DMEM in 96-well plates were hatched for 24 h and then cured with DON for 24 h. After 3 h incubation with 10 µL CCK-8 reagent in each well, absorption at 450 nm was determined on a plate photometer (Thermo Scientific, Waltham, MA, USA).

### 4.3. Hoechst 33,258 Staining

The impact of DON on the nuclear chromatin of cells was observed by Hoechst 33,258 staining. On sterile cover glasses, PHNCs were seeded and placed in 24-well plates for 24 h. The cells were then treated with the indicated concentrations of DON (0, 125, 250, 500, 1000, 2000 ng/mL) for 24 h. Cold PBS buffer was used to washed cells and fixed with 4% formaldehyde for 30 min, and then hatched with 100 µL Hoechst 33,258 staining solution for 5 min. After washing three times with PBS, the cells were viewed under a FV1000 laser confocal microscope (Olympus, Tokyo, Japan).

### 4.4. Determination of Apoptotic Cells

An annexin V-FITC/PI cell apoptosis assay kit (Wanleibio, Shenyang, China) was used to measure cell apoptosis. After treating for 24 h with different concentrations of DON, the cells were collected using two PBS washes. Then, 500 µL obligatory buffers were added to re-suspend the cell pellet, with the adding of 5 µL annexin V-binding and 5 µL propidium iodide (PI). Cells were stained for 15 min in the dark at room temperature, and then examined for apoptosis by a method called flow cytometry using 10,000 cells per sample (Becton Dickinson, Franklin Lakes, NJ, USA).

### 4.5. Detection of the Mitochondrial Membrane Potential (MMP)

JC-1 fluorescence kit (Thermo Scientific, Waltham, MA, USA) was used to measure mitochondrial membrane potential (MMP). PHNCs were treated with different concentrations of DON, ZIF, or 1000 ng/mL DON+ZIF for 24 h. JC-1 working solution was prepared in the proportion of 500 µL 1× incubation buffer to 1 µL JC-1. Cell pellets were re-suspended in 500 µL JC-1 operational solution and the cells were hatched for 15 min at 37 °C. The collected cells were washed two times with 500 µL 1× incubation buffer. Cells were then re-suspended in 500 µL 1× incubation buffer and evaluated by flow cytometry.

### 4.6. RT-PCR Analysis

In PHNCs, expression of Bcl-2 family members was detected by RT-PCR. The cells were treated with DON in different concentrations, ZIF, and 1000 ng/mL DON + ZIF for 24 h, respectively. Total RNA was isolated via Trizol (TaKaRa, Dalian, China), and cDNA was synthesized followed by a RT-PCR kit (Takara) rendering to the protocol of the manufacturer. The designed primers are listed in Table 1. PCR conditions were 95 °C for 1 min, followed by 40 cycles of 95 °C for 15 s, 58 °C for 20 s, and 72 °C for 20 s.

### 4.7. Western Blot Analysis

To determined protein levels of cleaved-caspase-3, cleaved-caspase-9, cytochrome C, and AIF Western blot analysis was used. DON was used in different concentration treat PHNC, ZIF, and 1000 ng/mL DON + ZIF. A mitochondria/cytosol fractionation kit (Beyotime Inst. Biotech, Beijing, China) was used to isolate mitochondrial and cytosolic proteins. Aliquots of 50 µg proteins were detached on a 12% SDS-polyacrylamide gel for cleaved-caspase-9, cleaved-caspase-3, and AIF, and on a 15% gel for cytochrome C, and then moved to PVDF membrane. Membranes were blocked with TBST buffer containing 5% bovine serum albumin and incubated with antibodies against β-actin (1:7500), AIF (1:200), CYCS (1:200), cleaved-caspase-3 (1:750), and cleaved-caspase-9 (1:750) at 4 °C overnight, followed by addition of horseradish peroxidase-linked anti-mouse/rabbit IgG. The bands were imagined via Super Signal West femto kit.

### 4.8. Immunofluorescence Analysis

The activity of caspase-3 was analyzed by immunocytofluorescence using laser confocal microscopy (FV100, Olympus, Tokyo, Japan). Briefly, PHNCs were grown-up on cover slips, and hatched with DON in different concentration, ZIF, or 1000 ng/mL DON + ZIF for 24 h and washed with cold PBS. After being static with 4% paraformaldehyde and with PBS washed thrice, the cells were permeabilized with 0.02% Triton X-100 for 3 min. Cells were blocked for 30 min at room temperature in 5% BSA, and hatched with cleaved-caspase-3 antibody at 4 °C for the whole night with secondary antibody (1:200) for 1 h. It was then washed thrice with PBS; nuclei were stained with DAPI for 3 min. Pictures were taken via Olympus FV10-ASW 1.7 Viewer software (Olympus, Tokyo, Japan).

### 4.9. Transcriptional Activity

The transcriptional activities of transcription factor AIF and P53 were determined by EMSA. A nuclear protein extraction kit (Sangon Biotech Co., Ltd., Shanghai, China) was used to isolate nuclear proteins. The concentrations of nuclear proteins were determined by Bio-Rad protein assay reagent (Sangon Biotech Co., Ltd., Shanghai, China). The binding reaction of extracted nuclear proteins (6 μg) with biotin-labeled probe was performed rendering to the protocol of the manufacturer. The complexes were detached by electrophoresis on non-denaturation 6% polyacrylamide Tris/borate/EDTA (TBE) gels and moved into a membranes made up of nylon. Membranes were crosslinked by a UV cross-linker (Cany Precision Instruments Co., Ltd., Shanghai, China) and the probe was noticed with the help of enhanced chemiluminescence solution (ECL; Pierce Biotechnology Inc., Chicago, IL, USA).

### 4.10. Statistical Analysis

All the data were expressed as mean ± standard deviation (SD). One-way ANOVA were used for statistical comparison, followed by Tukey’s post hoc test. *p* < 0.05 was measured statistically significant.

## Figures and Tables

**Figure 1 toxins-13-00073-f001:**
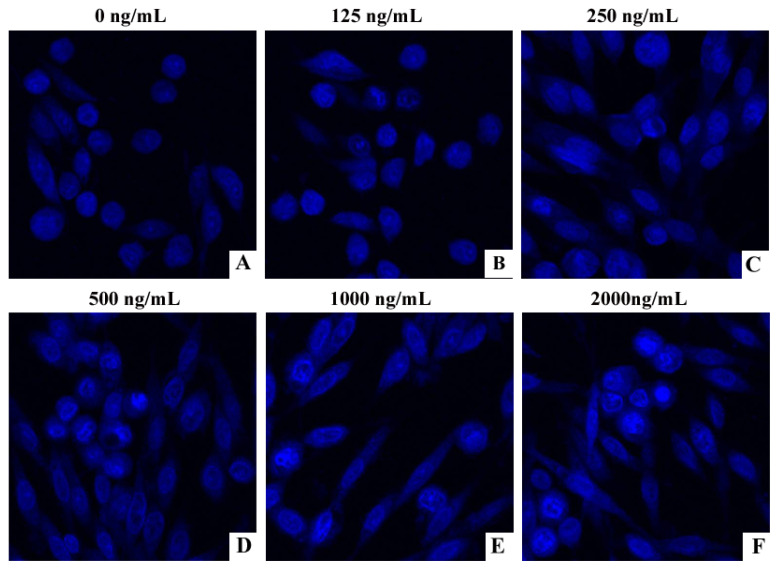
Apoptotic nuclear morphological changes highlighted by Hoechst 33,258 staining in cells treated with graded concentration of Deoxynivalenol (DON) (0–2000 ng/mL) for 24 h (800×). (**A**–**F**) indicates the effect of different concentrations of DON on the nucleus. The nuclei of DON-treated piglet hippocampal nerve cells (PHNCs) appeared bright blue. The intensity, proportion of nuclei appearing bright blue increased, and the morphology of the nucleus showed dose-dependent deformation with the increase of DON concentration.

**Figure 2 toxins-13-00073-f002:**
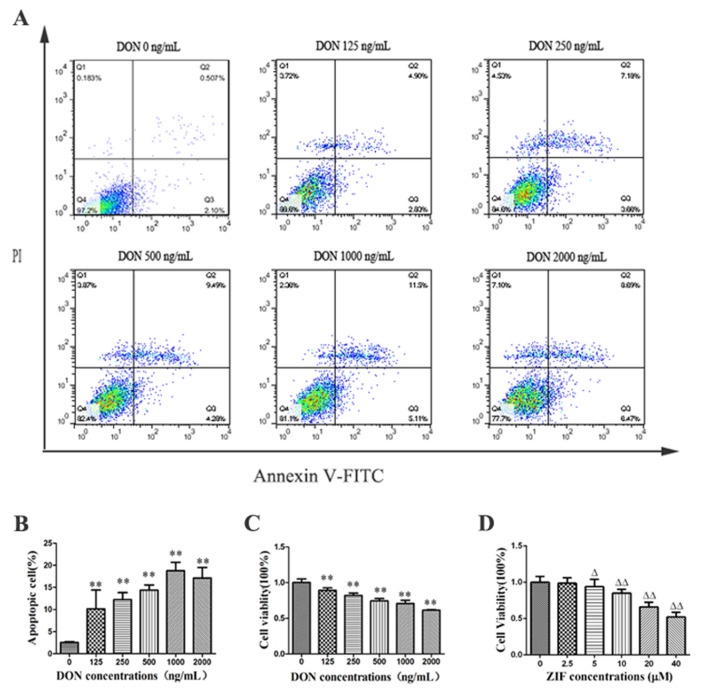
Effect of DON on PHNCs apoptosis rate and cells activity: (**A**) Flow cytometry of DON-treated PHNC apoptosis at different concentration; (**B**) The effect of different concentration of DON on PHNCs apoptosis rate, the apoptosis rate was increased with increasing concentrations of DON, and was highest at 1000 ng/mL; (**C**) The effect of different concentration of DON on PHNCs activity, cell activity was decreased with increasing concentrations of DON; (**D**)The effect of different concentration of Z-IETD-FMK (ZIF) on PHNCs activity, cell activity was decreased with increasing concentrations of ZIF. Data are presented as mean values ± SD (*n* = 3). ** DON group was highly significant different from controls (*p* < 0.01). Δ ZIF group was highly significant different from controls (*p* < 0.05). ΔΔ ZIF group was highly significant different from controls (*p* < 0.01).

**Figure 3 toxins-13-00073-f003:**
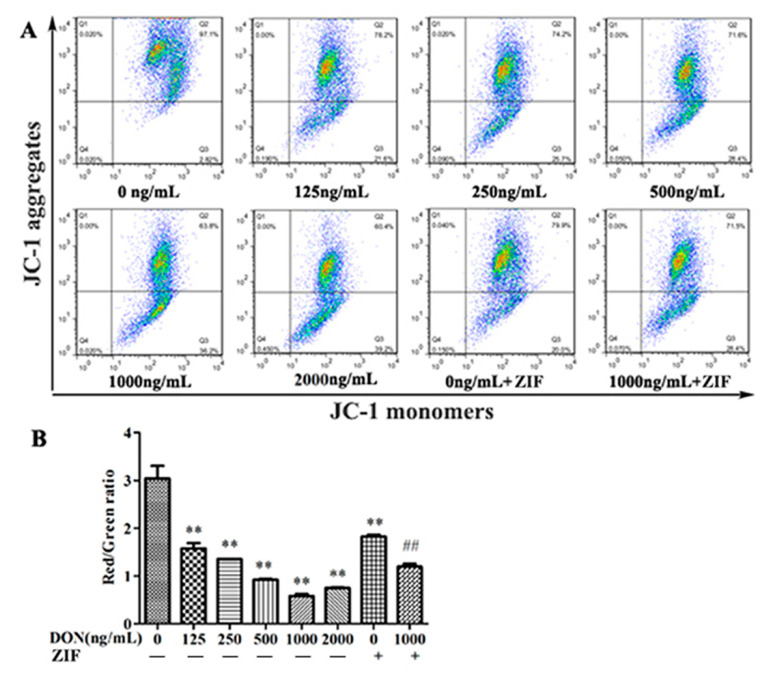
Effect of DON on mitochondrial membrane potentials in PHNC: (**A**) Different concentration of DON-treated PHNCs mitochondrial membrane potentials; (**B**) The effect of different concentration of DON on PHNCs mitochondrial membrane potentials. The MMP of PHNCs decreased with increasing concentrations of DON, caspase-8 inhibitor ZIF increased it. Data are mean values ± SD (*n* = 3). ** Highly significant difference vs. controls (*p* < 0.01). ## Highly significant difference vs. 1000 ng/mL DON group (*p* < 0.01).

**Figure 4 toxins-13-00073-f004:**
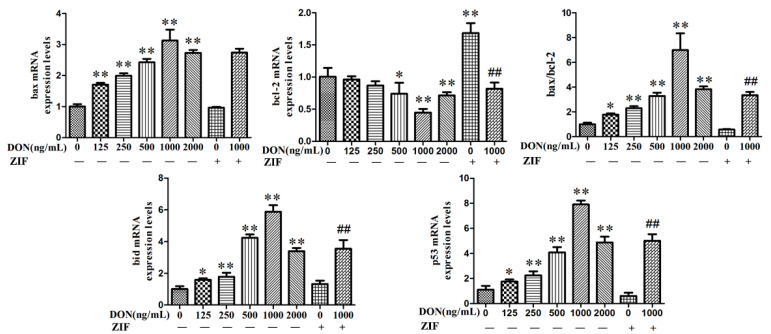
Effect of DON on apoptosis-related gene expression. The expression of *bcl-2* mRNA was decreased, and the expression of *bax*, *bid*, *p53* mRNA were dose-dependent increased with the increase of DON concentration, these effects were greatest when DON concentration was 1000 ng/mL. Caspase-8 inhibitor ZIF increased the expression of *bcl-2* mRNA and decreased the expression of *bax*, *bid*, *p53* mRNA. Data are mean values ± SD (*n* = 3). * Significant difference vs. controls (*p* < 0.05). ** Highly significant difference vs. controls (*p* < 0.01). ## Highly significant difference vs. 1000 ng/mL DON group (*p* < 0.01).

**Figure 5 toxins-13-00073-f005:**
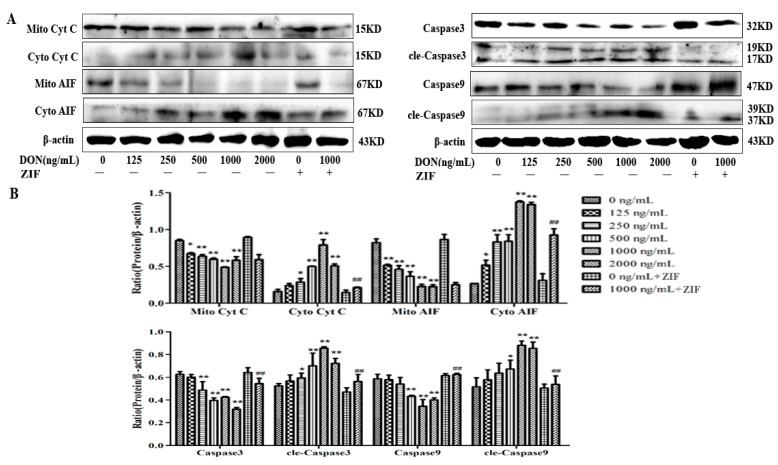
Effect of DON on apoptosis-related protein expression: (**A**) Western blot was used to detect proteins expression; (**B**) The effect of DON on Caspase3, Caspase9, cleaved-Caspase9(cle-Caspase9), cleaved-Caspase3(cle-Caspase3), Cyt C and AIF expression. The protein expression of Mito Cyt C, Mito AIF, Caspase 3, and Caspase 9 were decreased, and the protein expression of Cyto Cyt C, Cyto AIF, cle-Caspase3 and cle-Caspase9 were dose-dependent increased with the increase of DON concentration, these effects were greatest when DON concentration was 1000 ng/mL. Caspase-8 inhibitor ZIF increased the expression of Mito Cyt C, Mito AIF, Caspase 3, and Caspase 9 and decreased the expression of Cyto Cyt C, Cyto AIF, cle-Caspase3, and cle-Caspase9. Data are mean values ± SD (*n* = 3). * Significant difference vs. controls (*p* < 0.05). ** Highly significant difference vs. controls (*p* < 0.01). ## Highly significant difference vs. 1000 ng/mL DON group (*p* < 0.01).

**Figure 6 toxins-13-00073-f006:**
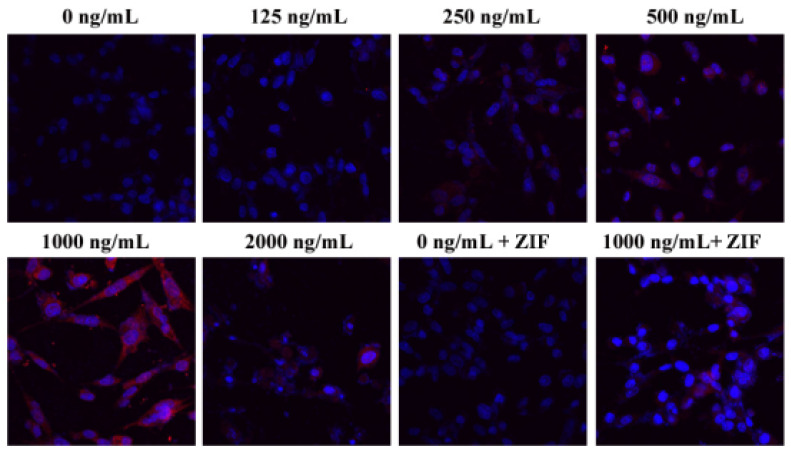
Effect of DON on Caspase3 activation (400×). Cells were treated for 24 h with different concentrations of DON (0, 125, 250, 500, 1000, 2000 ng/mL, ZIF, 1000 ng/mL DON+ZIF) and cleaved-caspase-3 was then measured, subjected to immunofluorescence analysis of cleaved-caspase3 activation(red), and nuclei were counterstained with DAPI (blue). When DON concentration was higher than 125 ng/mL, the expression of cleaved-caspase-3 was increased and the maximum expression was at 1000 ng/mL. Caspase-8 inhibitor ZIF decreased the expression of cleaved-caspase-3.

**Figure 7 toxins-13-00073-f007:**
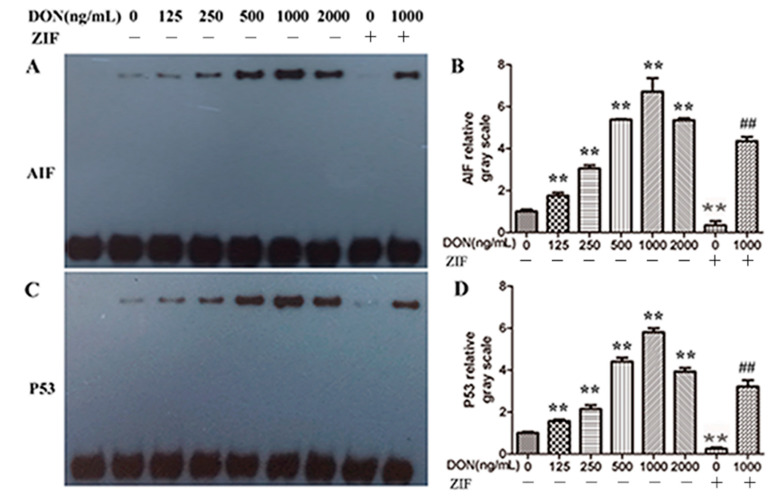
Effects of DON on the transcription activities of AIF and P53. Cells were treated for 24 h with different concentrations of DON (0, 125, 250, 500, 1000, 2000 ng/mL, ZIF, 1000 ng/mL DON + ZIF). Each treatment was replicated 3 times. Nuclear proteins were collected for the indicated time. (**A**,**B**): the electrophoretic mobility shift assay (EMSA) results of AIF; (**C**,**D**): the EMSA results of P53. DON was dose-dependent increased the transcriptional activities of AIF and P53 in PHNCs, these effects were greatest when DON concentration was 1000 ng/mL. Caspase-8 inhibitor ZIF decreased the transcriptional activities of AIF and P53. Data are mean values ± SD (*n* = 3). ** Highly significant difference vs. controls (*p* < 0.01). ## Highly significant difference vs. 1000 ng/mL DON group (*p* < 0.01).

**Table 1 toxins-13-00073-t001:** Parameters of primer for Bcl-2, Bax, Bid, P53, and GAPDH genes.

Gene	Accession Number	Primer	Sequences (5′→3′)	Product/bp
*GAPDH*	NM_002046	Forward	GGTGAAGGTCGGTGTGAACG	232
		Reverse	CTCGCTCCTGGAAGATGGTG	
*Bcl-2*	NC_000067.6	Forward	TGGGATGCCTTTGTGGAACT	153
		Reverse	GCAGGTTTGTCGACCTCACT	
*Bax*	NC_000073.6	Forward	GGTTTCATCCAGGATCGAGCA	151
		Reverse	TCCTCTGCAGCTCCATATTGC	
*Bid*	NM_197966.1	Forward	AGCTACACAGCTTGTGCCAT	186
		Reverse	CAGCTCGTCTTCGAGGTCTG	
P53	NC_000077.6	Forward	CCCAAACTGCTAGCTCCCAT	217
		Reverse	GGAGGATTGTGTCTCAGCCC	

## Data Availability

The datasets used and/or analyzed during the study are available from the corresponding author on reasonable request.

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
