# Peer review of "Deoxynivalenol Induces Caspase-8-Mediated Apoptosis through the Mitochondrial Pathway in Hippocampal Nerve Cells of Piglet"

_toxins, 2021, doi:10.3390/toxins13020073_

Round 1

Reviewer 1 Report

This paper assesses the interesting point on how DON can induce apoptosis on a nerve cell model. Even if the effects of DON are well known, is true that some clarification on its mechanisms of action are often missing. So, this paper brings some new key of understanding on the broad effects of DON.

I have minor comments.

- Graphical abstract: The legend of the arrows is missing. Some symbols are too small and not easy to read.

-Line 29, [1,2]: more recent references will be nice.

- I would recommend to change DON “pollution” by DON “contamination”.

- Section 2.1: Could you develop more this paragraph, with mentioning the doses (effect of DON seems to start only at 250ng/ml…).

- Figure 1: Could you describe more in detail the differences of morphology of the cells (the shape).

-L89: Could you indicate the meaning of “MMP” for the first use in the text.

-Figure 3: The quality of the figure is poor, numbers are not visible, and the concentration 500ng/ml is mentioned twice.

-Section 2.4: “p53 reduced with increasing concentrations of DON”, I don’t see that in the figure 4.

-Lines 112-113: Not clear, higher in comparison to what?

-Lines 130-131: I don’t really see a decrease for the Cyto AIF expression in Figure 5, or in comparison to what?

- Could you discuss the choice of the doses used?

- Line 175: “(schematic, figure 8)”, do you mean the graphical abstract?

- Figure 6: Have you quantified the immunofluorescence intensity? Maybe, precise at which dose the effect start.

- Why didn’t you look at the other caspases, caspases 2, 6, 7 for example?

Author Response

Dear Reviewer, 

On behalf of all the contributing authors, I would like to sincerely thank you for your response letter and for the reviewer’s constructive comments concerning our manuscript entitled “Deoxynivalenol Induces Caspase-8-Mediated Apoptosis Through the Mitochondrial Pathway in Hippocampal Nerve Cells of Piglet” (Manuscript ID: toxins-1044568). These comments are valuable and helpful for improving our manuscript. According to the reviewer’s comments, we have made substantial changes in several parts of the manuscript. In the revised version, all changes in the document have been highlighted in red. Point-by-point responses to the editor-in-chief and the three reviewers are listed below.

Point 1: Graphical abstract: The legend of the arrows is missing. Some symbols are too small and not easy to read.

Response 1: Thank you for your careful review. We have revised the graphical abstract.

Point 2: Line 29, [1,2]: more recent references will be nice.

Response 2: Thank you for your valuable comments. We have added recent reference in the text following your suggestion.

Point 3: I would recommend to change DON “pollution” by DON “contamination”.

Response 3: Thank you for your valuable comments. We have changed DON “pollution” by DON “contamination”.

Point 4: Section 2.1: Could you develop more this paragraph, with mentioning the doses (effect of DON seems to start only at 250ng/ml…).

Response 4: Thank you for your advice. We have reedited section 2.1.

Point 5: Figure 1: Could you describe more in detail the differences of morphology of the cells (the shape).

Response 5: Thank you for your advice. We have added more description in detail the differences of morphology of the cells.

Point 6: L89: Could you indicate the meaning of “MMP” for the first use in the text.

Response 6: Thank you for your careful review. We have added the full name of MMP in the text.

Point 7: Figure 3: The quality of the figure is poor, numbers are not visible, and the concentration 500ng/ml is mentioned twice.

Response 7: Thank you for your valuable comments. We have tried our best to provide the clearest picture. 500 ng/ml appears twice due to a marking error while drawing the picture; we have revised it, thank you for your correction.

Point 8: Section 2.4: “p53 reduced with increasing concentrations of DON”, I don’t see that in the figure 4.

Response 8: Thank you for your correction. We are sorry that we have made a mistake in writing the text. We have thoroughly revised it.

Point 9: Lines 112-113: Not clear, higher in comparison to what?

Response 9: Thank you for your careful review. We have revised it to make it clear in Lines 118-119.

Point 10: Lines 130-131: I don’t really see a decrease for the Cyto AIF expression in Figure 5, or in comparison to what?

Response 10: Thank you for your careful review. We have revised it in lines 136-137 .

Point 11: Could you discuss the choice of the doses used?

Response 11: Thank you for your valuable comments. We have explained it in lines 314-317.

Point 12: Line 175: “(schematic, figure 8)”, do you mean the graphical abstract?

Response 12: Yes. Thank for your careful review, we have revised it in Discussion.

Point 13: Figure 6: Have you quantified the immunofluorescence intensity? Maybe, precise at which dose the effect start.

Response 13: Thank you for your valuable comments. It can be clearly seen from Figure 6 that the expression of caspase 3 gradually increases when the concentration of DON exceeds 125 ng/ml and reaches the peak at 1000 ng/ml. FMK can effectively inhibit the expression of caspase 3 in PHNC cells.

Point 14: Why didn’t you look at the other caspases, caspases 2, 6, 7 for example?

Response 14: Thank you for your valuable comments. This paper studied that the apoptosis caused by mitochondrial pathway, A large amount of Cyt C was released into the cytoplasm induced by caspase 8, which activates caspase-9 in cytoplasm that further activates Caspase-3 and finally led to cell apoptosis in mitochondrial pathways. Caspase 3, 9, 8 have a cross-relation, which is more consistent with our research direction.

We believe that your suggestions have been very helpful in improving our manuscript, and hope that the explanations provided by us in the response letter will meet with your approval. We would like to sincerely thank you for your comments and valuable suggestions.

Sincerely,

Xichun Wang

Reviewer 2 Report

The manuscript “Deoxynivalenol Induces Caspase-8-Mediated 2 Apoptosis Through the Mitochondrial Pathway in 3 Hippocampal Nerve Cells of Piglet” has clear aims of the study, adequate laboratory methods and accurate conclusion. I found it fitting the scope and aims of the Journal.

I have several comments and suggestions for authors:

Introduction:

The apoptosis and its signalling pathways (the main topic) is not included in introduction.

I recommend re-writing this section to be more relevant with the findings.

Results:

The results are clearly describe and laboratory methods are adequate.

Some figures (2B, 3A, 3B, 4, 5B, 7B, 7D) have insufficient graphical quality. Please improve it.

Abbreviation are not explain when first used in the text (for example MMP line 89, FMK line 94, …).

Discussion:

The discussion is extensive but contains lot of parts that should be better in the introduction.

The authors use porcine cell line but I miss comparison with other studies done with porcine cells. I recommend add some references.

Author Response

Dear Reviewer, 

On behalf of all the contributing authors, I would like to sincerely thank you for your response letter and for the reviewer’s constructive comments concerning our manuscript entitled “Deoxynivalenol Induces Caspase-8-Mediated Apoptosis Through the Mitochondrial Pathway in Hippocampal Nerve Cells of Piglet” (Manuscript ID: toxins-1044568). These comments are valuable and helpful for improving our manuscript. According to the reviewer’s comments, we have made substantial changes in several parts of the manuscript. In the revised version, all changes in the document have been highlighted in red. Point-by-point responses to the editor-in-chief and the three reviewers are listed below.

Point 1: Introduction:

The apoptosis and its signaling pathways (the main topic) is not included in introduction.

I recommend re-writing this section to be more relevant with the findings.

Response 1: Thank you for your valuable comments. We have revised Introduction, and added introduction about apoptosis and signaling pathways in lines 56-62.

Point 2: Results:

The results are clearly describe and laboratory methods are adequate.

Some figures (2B, 3A, 3B, 4, 5B, 7B, 7D) have insufficient graphical quality. Please improve it.

Abbreviation are not explain when first used in the text (for example MMP line 89, FMK line 94, …).

Response 2: Thank you for your careful review. We have tried our best to provide the clearest picture, and we have explained the abbreviation when it first used in the text in our revised manuscript.

Point 3: Discussion:

The discussion is extensive but contains lot of parts that should be better in the introduction.

The authors use porcine cell line but I miss comparison with other studies done with porcine cells. I recommend add some references.

Response 3: Thank you for your valuable comments. We have added some references about other studies done with porcine cells, which are more related with our study.

We believe that your suggestions have been very helpful in improving our manuscript, and hope that the explanations provided by us in the response letter will meet with your approval. We would like to sincerely thank you for your comments and valuable suggestions.

Sincerely,

Xichun Wang

Reviewer 3 Report

Title: Deoxynivalenol Induces Caspase-8-Mediated  Apoptosis Through the Mitochondrial Pathway in  Hippocampal Nerve Cells of Piglet

The authors have addressed the induction of apoptosis by deoxynivalenol (Don) in Piglet hippocampal nerve cells (PHNCs). They have used different methods to investigate the effect including, flowcytometry, cell viability assays, immunofluorescense antibody test, western blot and EMSA to investigate the responses induces and describe the toxic effect. The main conclusion was that apoptosis in induced through the mitochondrial associated caspase 8 pathway. The findings are interesting but there are different issues that need to be addressed before the paper is suitable for publication.

Main concerns:

  • Over all, the manuscript is poorly written and need to be improved significantly.
  • The results are not presented in proper and detailed way. I will provide the first results as an example to guide the author to properly describe the results.

The authors have written “Laser confocal microscopy was used to detect nuclear features (Figure 1). Untreated (control) PHNCs showed uniformly blue nuclei (Figure 1 A). However, different concentrations of DON induced significant nuclear changes. The nuclei of DON-treated PHNCs appeared bright blue, and the intensity and proportion of nuclei appearing bright blue increased as DON concentrations increased (Figure 1 B-F)

Here what is described as bright blue is chromatin condensation. The results should explain what have been observed , at what concentrations and the differences observed in the different concentrations (i.e. are the chromatin condensation or nuclear change are increasing/decreasing with the concentrations? What are the differences between the different concentrations?).

Hence, all the results should be explained in detail not only using general statements.

  • The figure text should be self-explanatory and should provide enough details that allow the reader to understand the results presented. This is not the case for almost all the figures.
  • The Hoechst staining (Fig. 1) show little DNA fragmentation while in fig.2C the data show a very significant reduction in cell viability, please explain?
  • Fig 2A: The number of cells in the 0 ng/ml concentration seems less than the others. Please explain?
  • The FMK treatment has been suddenly appeared in the results without giving the rational of using it as a control. The chemical is not described properly, and I assume it is z-vad-fmk, if so this is a pan caspase inhibitor not caspase 8 inhibitor as described in line 78.
  • Moreover, the FMK treatment has resulted in significant decrease in the MMP and this may indicate that the concentrations used are high and may have induced some toxic effect. Have the authors tested different doses and assessed toxicity?
  • Fig.5, Please provide quantification of cleaved and non-cleaved caspase 3 and 9 for a better illustration of the effect.
  • The discussion is lengthy but poorly written in my opinion. It should include the toxicity effects of DON in compare to similar toxins and also compare the apoptotic and mitochondrial changes observed in this study compared to previous findings. Comparing the effect seen on the different type of cells or cell lines will also be important.

Minor comment:

  • Full name should always be presented before the abbreviation is used such as in line 7 for fusarium and MMP in line 89.
  • Please provide a proper description for the x and y accesses in fig. 3A, not HL 1 and HL 2
  • The introduction is lacking references in some places such as in line 35 and line 36.

Author Response

Dear Reviewer, 

On behalf of all the contributing authors, I would like to sincerely thank you for your response letter and for the reviewer’s constructive comments concerning our manuscript entitled “Deoxynivalenol Induces Caspase-8-Mediated Apoptosis Through the Mitochondrial Pathway in Hippocampal Nerve Cells of Piglet” (Manuscript ID: toxins-1044568). These comments are valuable and helpful for improving our manuscript. According to the reviewer’s comments, we have made substantial changes in several parts of the manuscript. In the revised version, all changes in the document have been highlighted in red. Point-by-point responses to the editor-in-chief and the three reviewers are listed below.

Point 1: Over all, the manuscript is poorly written and need to be improved significantly.

Response 1: Thank you for your comment. We have thoroughly revised our manuscript and made substantial changes where necessary or suggested by the reviewers. We also checked the entire manuscript for any grammatical or any other errors with the help of an English-language professor to correct all the mistakes.

Point 2: The results are not presented in proper and detailed way. I will provide the first results as an example to guide the author to properly describe the results.

Response 2: Thank you for your valuable comment and guidance regarding our results. We have revised the results section following your suggestion.

Point 3: The figure text should be self-explanatory and should provide enough details that allow the reader to understand the results presented. This is not the case for almost all the figures.

Response 3: Thank you for your valuable comments. We have checked and revised the figure text.

Point 4: The Hoechst staining (Fig. 1) show little DNA fragmentation while in fig.2C the data show a very significant reduction in cell viability, please explain?

Response 4: Thank you for your valuable comments. In figure 1, when the concentration of DON was greater than 125 ng/ml, nuclei appeared small and bright blue, this phenomenon indicated that apoptosis has occurred. Therefore, the results of figure 1 and figure 2C are correspond to each other.

Point 5: Fig 2A: The number of cells in the 0 ng/ml concentration seems less than the others. Please explain?

Response 5: Thank you for your comment. The number of cells detected by the flow cytometer we used was 10000 at a time, the number of cells in the 0 ng/ml concentration seems less than the others because the parameters were not adjusted well when we drawing the figure. We have revised figure 2A to dispel this illusion.

Point 6: The FMK treatment has been suddenly appeared in the results without giving the rational of using it as a control. The chemical is not described properly, and I assume it is z-vad-fmk, if so this is a pan caspase inhibitor not caspase 8 inhibitor as described in line 78.

Response 6: Thank you for your valuable comments. The FMK we used in our study was Z-IETD-FMK. We have given the dose and full name of FMK in the text.

Point 7: Moreover, the FMK treatment has resulted in significant decrease in the MMP and this may indicate that the concentrations used are high and may have induced some toxic effect. Have the authors tested different doses and assessed toxicity?

选择

Response 7: Thank you for your valuable comments. Z-IETD-FMK is a common caspase 8 inhibitor. We screened the concentration of it at the beginning of the experiment, and finally determined the dose in this text by combining with the results of other references. It is not difficult to find from Figure 3, 6 and 7 that the concentration we chose is appropriate, which can completely achieve our experimental purpose.

Point 8: Fig.5, Please provide quantification of cleaved and non-cleaved caspase 3 and 9 for a better illustration of the effect.

Response 8: Thank you for your valuable comment. We have provided relevant histogram in figure 5B.

Point 9: The discussion is lengthy but poorly written in my opinion. It should include the toxicity effects of DON in compare to similar toxins and also compare the apoptotic and mitochondrial changes observed in this study compared to previous findings. Comparing the effect seen on the different type of cells or cell lines will also be important.

Response 9: Thank you for your valuable comment. We have revised the Discussion following your suggestion.

Point 10: Full name should always be presented before the abbreviation is used such as in line 7 for fusarium and MMP in line 89.

Response 10: Thank you for your careful review. We have revised and provide full name for each abbreviation in our revised paper.

Point 11: Please provide a proper description for the x and y accesses in fig. 3A, not HL 1 and HL 2

Response 11: Thank you for your careful review. We have revised it.

Point 12: The introduction is lacking references in some places such as in line 35 and line 36.

Response 12: Thank you for your valuable comments. We have added relevant references.

We believe that your suggestions have been very helpful in improving our manuscript, and hope that the explanations provided by us in the response letter will meet with your approval. We would like to sincerely thank you for your comments and valuable suggestions.

Round 2

Reviewer 3 Report

I appreciate the author's efforts to improve the manuscript and follow the suggestions. However, although the authors have addressed some of the issue raised, most of the issues remain unchanged.

  • The manuscript is still poorly written and need extensive language editing.
  • Introduction still lacking references. Firm statement such as the sentences from line 35-38 need to be supported by references.
  • The figure texts has been improved but still need additional editing.
  • The discussion has not been improved.
  • Z-IETD-FMK need to be added to the figure text and to be given a better abbreviation than FMK. In addition data about the toxicity and testing different concentration of this inhibitor should be added.

Author Response

Point 1: The manuscript is still poorly written and need extensive language editing.

Response 1:Thank you for your valuable comment. This manuscript has been extensively revised by an English-speaking specialist lived in America.

Point 2: Introduction still lacking references. Firm statement such as the sentences from line 35-38 need to be supported by references.

Response 2: Thank you for your valuable comment. We have added three references in Line 35-38 and Line 48.

Point 3: The figure texts has been improved but still need additional editing.

Response 3: Thank you for your valuable comment. We have added relevant descriptions in the legend of figure 1 - figure 7.

Point 4: The discussion has not been improved.

Response 4: We are terribly sorry about that, we have revised the Discussion thoroughly this time.

Point 5: Z-IETD-FMK need to be added to the figure text and to be given a better abbreviation than FMK. In addition data about the toxicity and testing different concentration of this inhibitor should be added.

Response 5: Thank you for your valuable suggustion. We have abbreviated Z-IETD-FMK to ZIF and replaced the FMK in the text and figures, and added the data about the toxicity and testing different concentration of ZIF in figure 2D.